# Femtosecond-precision electronic clock distribution in CMOS chips by injecting frequency comb-extracted photocurrent pulses

Minji Hyun[1,3], Hayun Chung [2,3] ✉, Woongdae Na[2] & Jungwon Kim [1] ✉

A clock distribution network (CDN) is a ubiquitous on-chip element that provides synchronized clock signals to all different circuit blocks in the chip. To maximize the chip performance, today's CDN demands lower jitter, skew, and heat dissipation. Conventionally, on-chip clock signals have been distributed in the electric voltage domain, resulting in increased jitter, skew, and heat dissipation due to clock drivers. While low-jitter optical pulses have been locally injected in the chip, research on effective distribution of such high-quality clock signals has been relatively sparse. Here, we demonstrate femtosecond-precision distribution of electronic clocks using driver-less CDNs injected by photocurrent pulses extracted from an optical frequency comb source. Femtosecond-level on-chip jitter and skew can be achieved for gigahertz-rate clocking of CMOS chips by combining ultralow comb-jitter, multiple driver-less metal-meshes, and active skew control. This work shows the potential of optical frequency combs for distributing high-quality clock signals inside high-performance integrated circuits, including 3D integrated circuits.

In a high-performance CMOS chip, well synchronized clock signals must be provided to all circuit elements by the CDN[1–3]. Conventionally, on-chip clock signals have been generated and distributed via tree- or mesh-based networks in the electric voltage domain[1–5]. As electronic clock generators are unable to drive the clock load directly, many levels of clock drivers have been employed to overcome the bandwidth limitations. Due to the large number of clock drivers in processors and digital systems, substantial power consumption occurs, accounting for 10–30% of overall chip power consumption[6–9] and causing on-chip thermal problems. As each clock driver adds timing uncertainty, jitter (random variation in clock arrival time) and skew (spatial variation in clock arrival time) performances are also significantly impaired. As a result, both jitter and skew performances have been limited to several to tens ps range[1,5,8,10]. While the demand for

higher data rates necessitates a tighter jitter and skew budget, clock skew and jitter deteriorate as on-chip process, voltage and temperature (PVT) variations worsen in deep sub-micron processes.

Optical pulse trains generated from femtosecond mode-locked lasers and optical frequency combs can provide intrinsically low timing jitter down to the sub-100-attoseconds regime[11,12]. For this reason, they have been actively used for precise timing applications, including synchronization of large-scale X-ray free-electron lasers[13], high-speed photonic analogue-to-digital converters (ADCs)[14] and photonics-based radars[15], to name a few. Naturally, utilizing ultralow-jitter optical pulses in electronic or electronic-photonic chips has also been actively investigated[16–23]. For the on-chip clocking, an early-stage pioneering experimental study employed ultrashort optical pulses generated from an 80 MHz mode-locked Ti:sapphire laser to inject a local clock

[1]Korea Advanced Institute of Science and Technology (KAIST), Daejeon 34141, Korea. [2]Korea University, Sejong 30019, Korea. [3]These authors contributed equally: Minji Hyun, Hayun Chung. ✉e-mail: hcchung@korea.ac.kr; jungwon.kim@kaist.ac.kr

signal to a four-stage flip-flop with 5.9-ps timing jitter[16], and later to a multiplexer down to 0.93-ps jitter[24]. Directly injecting low-jitter optical pulses in the chip for optical sampling and ADCs was also investigated, showing that local pulse injection with tens-femtosecond-level relative jitter (which is inferred from the measured signal-to-noise ratio of a converter) is possible[21,23]. While ultralow-jitter local clocks could have been injected and generated in the chip, to our knowledge, there has been little study on the distribution of such clock signals with femtosecond precision inside the chip.

Here, we demonstrate femtosecond-precision distribution of electronic clock signals inside a CMOS chip using a driver-less CDN injected by photocurrent pulses. Our approach directly drives driver-less metal-mesh sections in a CMOS chip by low-jitter, sharp-edged photocurrent pulses extracted from balanced photodiodes when illuminating them with optical pulse trains generated from an optical frequency comb source. On the one hand, the use of low-jitter and ultrashort optical pulses[11,12] generated from an optical frequency comb source can naturally reduce the timing jitter of the extracted electronic clock signals in the CMOS chip. On the other hand, the use of ultra-short optical pulses for generating strong photocurrent pulses can remove the necessity of clock drivers in the CDN and, therefore, can significantly reduce the skew and on-chip heat dissipation. By dividing the clock domains into multiple sections with driver-less metal structures, the CDN can be implemented in a scalable way. The final inter-domain skews can be compensated by monitoring them with a ~100-fs-resolution on-chip time-to-digital converter (TDC)[25] and sending the timing information to the optical domain.

## Results

### Concept and experimental setup

Figure 1a shows the conceptual schematic diagram of the demonstrated photonic CDN. The basic idea is to generate a clock signal by charging and discharging the capacitive load (the global metal-mesh in the CMOS chip $C_{CDN}$ in Fig. 1a) by short photocurrent pulses with opposite polarity. While a demonstration of injecting photocurrent pulses to a local capacitive load (with ~100 fF load[26]) has been shown[16], our approach is different in that it injects the current pulses directly into the global metal-mesh in the CMOS chip. The optical pulse train, with pulse repetition rate equivalent to the targeted clock rate, is split into two by a coupler. To create a 50% duty-cycle clock, one path experiences a delay corresponding to half of the clock period (T/2 in Fig. 1a). The set of optical pulse trains is applied to an off-chip balanced photodiode. Variable optical attenuators (VOAs in Fig. 1a) are inserted to compensate for the mismatch in photogenerated carriers caused by process variation of photodiodes, waveguide losses, and coupling ratio imbalance in the optical coupler. The balanced photodiode converts optical pulses to low-jitter, high-peak, and short photo-current pulses with opposite polarity, which can directly charge and discharge the capacitance imposed by clock distribution structure and clock loads, generating lower-jitter and sharp clock edges in the vol-tage domain. Furthermore, by putting multiple metal-mesh sections on a chip, it is possible to achieve not only scalability in drivable load but also efficient operation of different circuit blocks under various clock load conditions.

Figure 1b shows the schematic of the photonic CDN demonstra-tion experiment. A mode-locked Er-fibre oscillator with a 250 MHz repetition rate is employed as an optical frequency comb source. By interleaving the 250 MHz pulses with a two-stage fibre Mach-Zehnder interferometer-based pulse interleaver[27], we were able to generate 500 MHz and 1 GHz optical pulse trains. We also used a separate 2 GHz Er-waveguide laser to generate a clock signal up to 2 GHz rate. A test chip prototype is designed and fabricated in 65 nm CMOS process (the chip die photo shown in Fig. 1b). The test chip includes three driver-less clock distribution structures as a form of metal mesh, clamper circuits for protection, D-flipflop-based active loads, an H-tree-based CDN for comparison, and a sub-ps-resolution stochastic TDC[25] to measure skew (see Methods). Note that, in order to demonstrate scalability in drivable load and efficient operation of different circuit blocks with different clock load conditions, we designed the chip with three separate clock domains (labelled as C0, C1 and C2 in Fig. 1b). Clock domains C1 and C2 consist of additional dynamic clock loads (i.e., 6036 and 3360 D-flipflops, respectively) to study the impact of active loads. In addition, for comparison with conventional electronic CDN, an H-tree structure with 23 stages of clock drivers (1194 in total) is

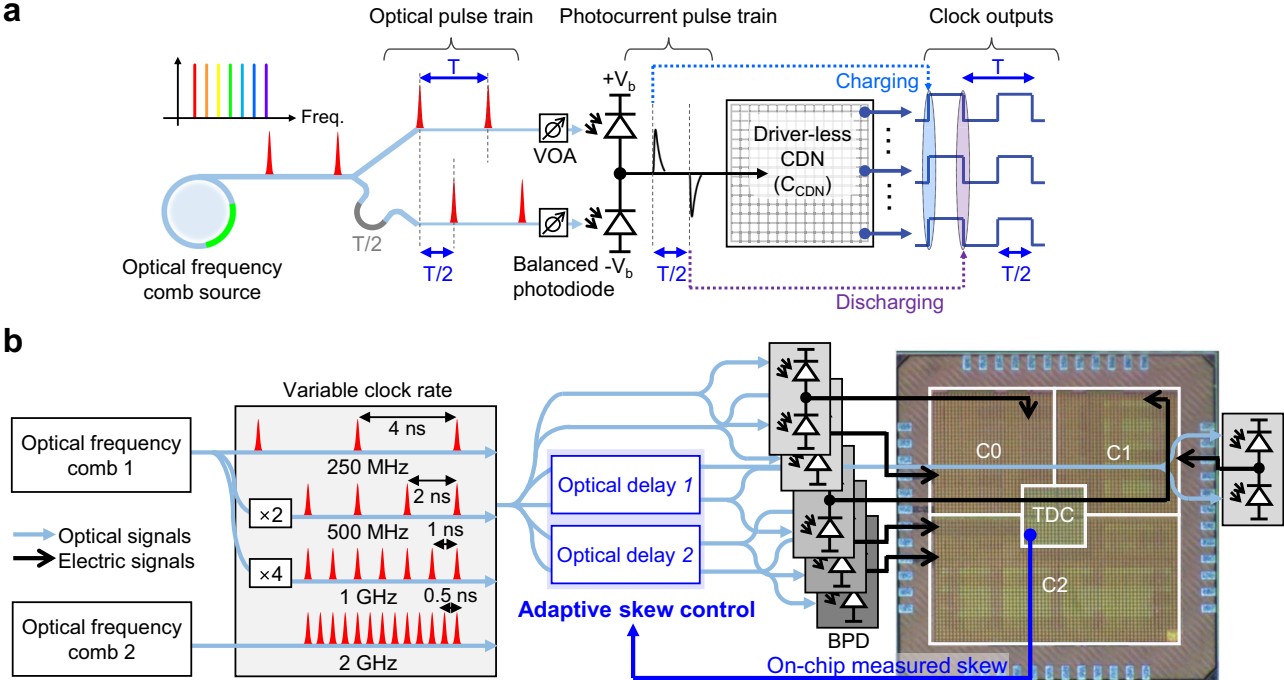

**Fig. 1 | Photonic CDN with an optical frequency comb as a master clock. a** Conceptual schematic diagram of the demonstrated photonic CDN. VOA Variable optical attenuator. **b** Schematic of the photonic CDN demonstration experiment. BPD Balanced photodiode, TDC Time-to-digital converter.

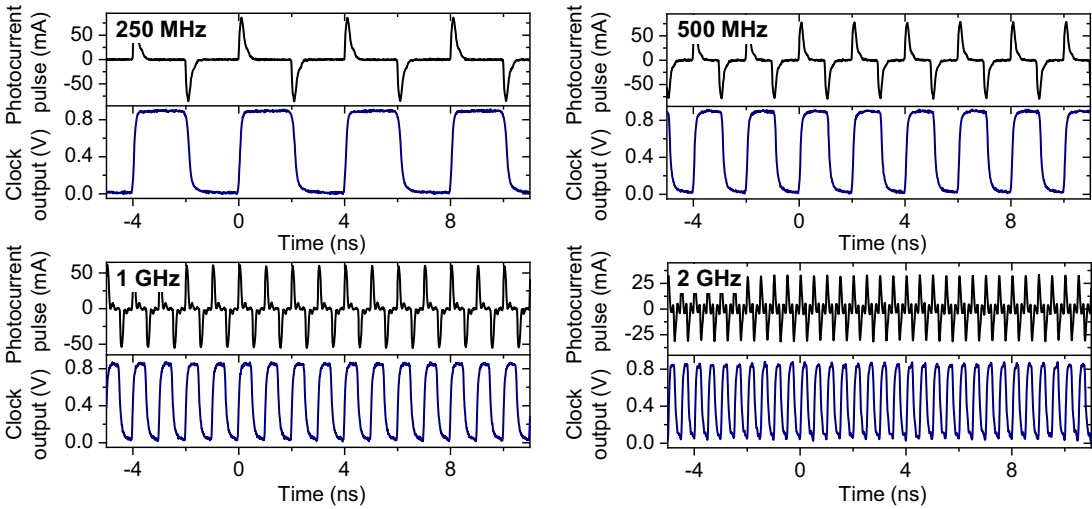

**Fig. 2 | Resulting waveforms at different clock rates.** Photocurrent pulse waveforms from the BPD (top panels) and output digital clock signal waveforms (bottom panels), which were measured independently, at clock rates of 250 MHz, 500 MHz, 1 GHz, and 2 GHz. Clock signal waveforms for 250 MHz, 500 MHz, and 1 GHz are the clock of the C2 clock domain when dynamic clock loads and H-tree structure CDN are running. The 2 GHz clock signal waveform is measured for the C0 clock domain.

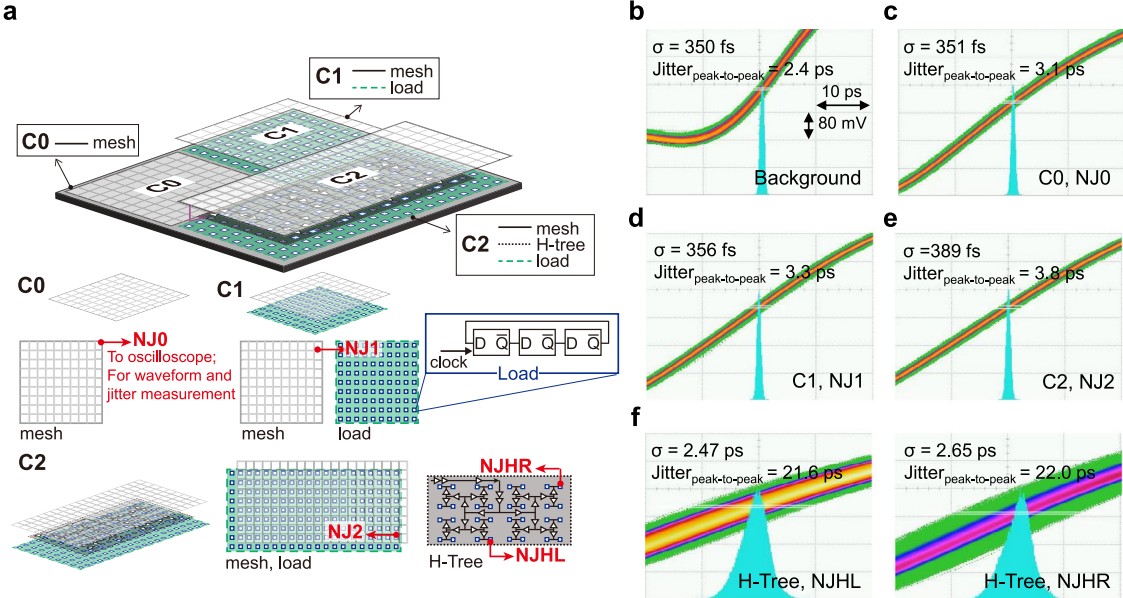

**Fig. 3 | Clock jitter measurement results. a** Diagram of the test chip related to jitter measurement. Note that the output clock signal from C2 is used as the input clock signal of the H-tree. **b** Jitter measurement background. **c–f** Jitter histograms of the generated 1 GHz clocks were measured off-chip through an open-drain clock driver in the test chip. Standard deviation and peak-to-peak jitter values were extracted from 100,000-hit-count histograms of the signal crossing at half maximum level.

placed under the C2 structure. The chip area is 2 mm by 2 mm, and the capacitance of each clock mesh plus clock load is 12.7 pF, 17.0 pF and 26.6 pF for C0, C1, and C2 clock domains, respectively, which are estimated with the parasitic extraction tool[28]. Note that adaptive skew control between different clock domains can be also achieved by monitoring the inter-domain skew with the on-chip TDC and transmitting this information to the delay control actuator in the optical path.

By applying optical pulses to an off-chip 5-GHz balanced photodiode connected to the pad, current pulses are directly injected into each clock domain on the chip. Note that, to drive a larger clock load (for example, C2 in Fig. 1b), multiple balanced photodiodes can be connected in parallel to increase the effective charges of photocurrent pulses. The on-chip clocks are fed out with one clock driver followed by an open drain driver, while the clocks from the H-tree CDN are fed

out with two stages of clock driver followed by an open drain driver. Figure 2 shows the measured waveforms of the input photocurrent pulse trains and the generated CDN clock outputs when 250 MHz, 500 MHz, and 1 GHz optical pulse trains are applied to the C2 clock domain when the dynamic clock load as well as the H-tree section are turned on (clock outputs with different clock rates and load conditions are presented in Supplementary Fig. 1). The clock output for 2 GHz optical pulse train is also generated from the C0 clock domain.

## Jitter performance

Figure 3a shows the system configuration showing the locations and elements related to the jitter measurement. The on-chip clock at each clock domain is fed out from nodes denoted as red arrows in Fig. 3a with an open-drain clock driver for off-chip waveform and jitter measurement using an oscilloscope (see Methods). Note that the trigger

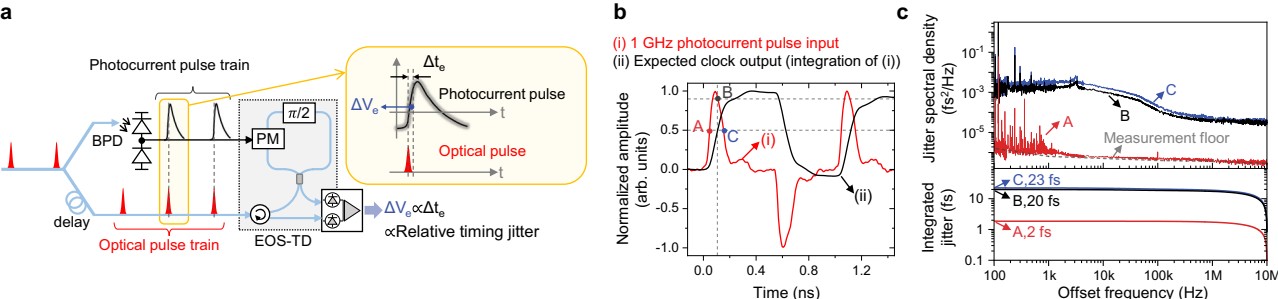

**Fig. 4 | Photocurrent pulse jitter characterization. a** Experimental setup to characterize the timing jitter of photocurrent pulses at different temporal positions using the EOS-TD[29]. **b** (i) Measured photocurrent pulse waveform and (ii) integrated clock waveforms. **c** Measured timing jitter power spectral densities at different locations (corresponds to points A, B, and C in Fig. 4b) of photocurrent pulses.

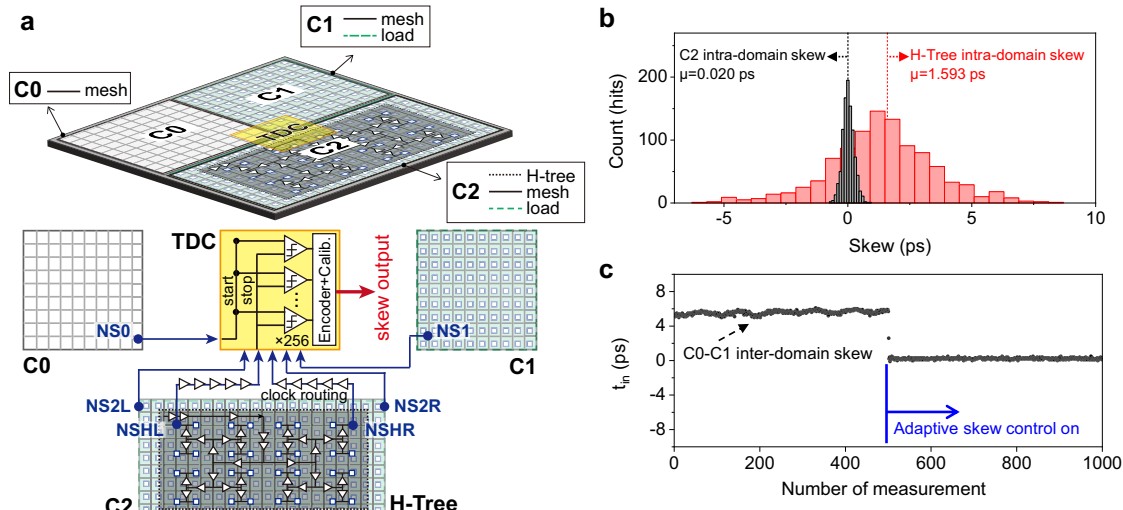

**Fig. 5 | Skew measurement results. a** System diagram of the test chip related to skew measurement. **b** Histogram of intra-domain skew for C2 and H-tree. **c** 1000-consecutive measured inter-domain skew between C0 and C1. Slowly-varying inter-domain skew offset of C1 is compensated by adaptive skew control.

signal for the oscilloscope is provided by the photodetected signal of the optical pulse train, and the rms jitter measurement background is determined to be ~350 fs rms (Fig. 3b). Since the edge of photocurrent pulses, which has fs-level jitter[29], was used as the trigger signal (see Methods), we attribute this ~350-fs-level background jitter to the instrument limit of the used oscilloscope[30]. The jitter measurement results of 1 GHz digital clock distributed through the photonic CDN and the conventional H-tree CDN are presented in Fig. 3c–f. The clock signals extracted from the C0, C1 and C2 clock domains show rms jitter of 351 fs, 356 fs, and 389 fs, respectively (Fig. 3c–e), which implies that the current jitter measurements are limited by the instrument noise of the used oscilloscope and the actual jitter in the photonic CDN is much lower. Any noticeable difference in timing jitter is not observed according to various clock load conditions within the same clock domain (see Supplementary Fig. 2). On the other hand, the conventional H-tree CDN shows significantly higher rms timing jitter of ~2.5 ps (Fig. 3f).

Because the frequency comb's timing jitter and the optical-to-electronic conversion jitter are both known to be significantly lower, the CDN jitter directly from the clock mesh might also be much lower than the oscilloscope's instrument limit (350 fs). To investigate the actual jitter, we employed the electro-optic sampling-based timing detector[29] to characterize the timing jitter at different temporal positions of photocurrent pulses (Fig. 4a, see Methods). By locating the corresponding temporal position where the integrated clock waveform reaches the 50% level (Fig. 4b), we can infer the timing jitter of the

generated clock signal. The measured result shows that the expected rms timing jitter at 50% level is ~20 fs (Fig. 4c). Note that this jitter level (when using a 5 GHz photodiode) is higher than the result when using a 12 GHz or 22 GHz photodiodes[29], and a higher bandwidth photodiode can reduce timing jitter to well below 10 fs.

## Skew performance

As shown in Fig. 5a, the skew is assessed by an on-chip stochastic TDC with ~100-fs resolution[25] (see Methods). The TDC measures the clock arrival time difference between the clock edges at node NS0 and at other nodes (denoted as NS1, NS2L, NS2R, NSHL and NSHR in Fig. 5a), and as a result, can determine the inter-domain skew. For intra-domain skew, locations of left and right clock sinks denoted as blue circles in Fig. 5a (for example, NS2L and NS2R for C2) were chosen to represent the worst-case skew, which can be obtained by subtracting the arrival times between these two points. To calibrate the TDC, characteristic curves (digital code output versus input time) are measured for each 1 GHz clock signal (see Methods). Subsequently, the TDC measures 1000-consecutive skews at the mid-point time offset of the curve, and the histogram and sequential data are presented in Fig. 5b, c, respectively. As shown in Fig. 5b, the extracted intra-domain mean skew of C2 (photonic CDN) is only 20 fs, whereas the H-tree mean skew is 1.6 ps. Note that this H-tree is used to compare to typical CDNs with clock drivers, and that there are a number of approaches[31–33] for electronic CDNs that have reduced skew such as length matching[31] and active deskewing[32].

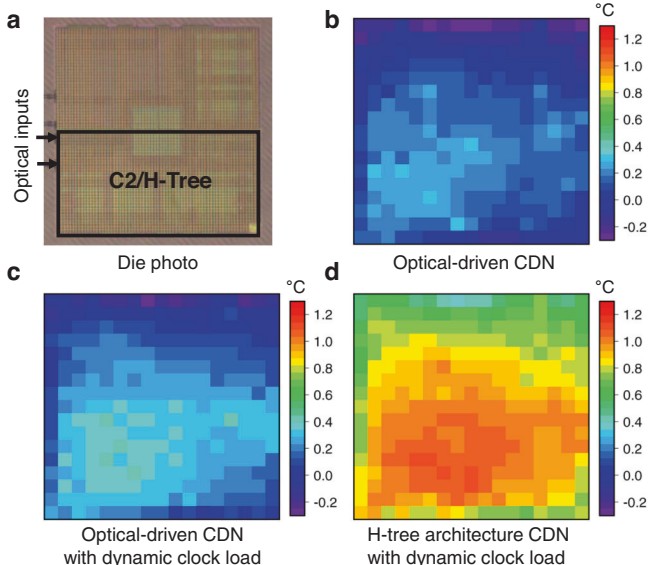

**Fig. 6 | Die thermal map. a** Die photo. The operated partial region for thermal analysis is shown as a solid box. **b** Temperature rise compared to bare die when optical input is entered to generate digital clock through proposed optical CDN. A maximum and mean temperature rise of 0.3 °C and 0.1 °C have been observed. **c** Thermal image when optical CDN is operated with dynamic clock load. A maximum and mean temperature rise are 0.4 °C and 0.2 °C, respectively. **d** Thermal image when H-tree CDN is driven. A maximum and mean temperature rise are 1.1 °C and 0.9 °C, respectively.

Another attractive feature of our approach is that adaptive skew control between different clock domains can be achieved by monitoring the inter-domain skew with the TDC and using this information to control the optical delay. For the demonstration experiment, a slowly varying delay offset is introduced to the optical path to the C1. The output from the TDC with skew information between C0 and C1 feedback-controls an actuator in the optical path connected to one of the two clock domains. Figure 5c shows that the adaptive skew control is successfully demonstrated. The resulting skew between C0 and C1 clock domains is measured to be 231 fs (Fig. 5c). The inter-domain skew control performance is currently limited by long skew-data acquisition time caused by the long off-chip communication time between several instruments such as logic analyzer, pattern generator, and computer (see Supplementary Fig. 3), which are used to operate the on-chip TDC. With improved acquisition speed by integrating the functions in same chip or utilizing a field programmable gate array (FPGA), inter-domain skew well below 100 fs may be possible.

### On-chip CDN power consumption and thermal imaging results

By eliminating clock drivers, the on-chip power consumption and heat dissipation from the CDN clock domains can be significantly decreased. Here, we compare the cases of C2 and H-tree, which share the same area on the test chip. We first assessed the overall power consumption and CDN power consumption when driven by the H-tree CDN. The measured overall power consumption and CDN power consumption were 51.5 mW and 22.1 mW, respectively, which shows that the relative ratio of CDN power consumption was ~43% for the test chip. In contrast, the on-chip power dissipation for the driver-less distribution in C2 was calculated as ~112 μW for 1 GHz clocking, which is significantly lower than the H-tree case, by using the extracted resistance of the metal grid of C2 (0.16 ohm) and the measured current pulse waveforms. To assess the thermal loading by clock distribution, thermal images were taken during the operation of C2 and H-tree (Fig. 6). When the optical input is used to drive the photonic CDN (Fig. 6b), the maximum and mean temperature rises by 0.3 °C and

0.1 °C, respectively; in contrast, when the H-tree structure CDN is used, the maximum and mean temperature rises by 1.1 °C and 0.9 °C, respectively (Fig. 6d). Note that, compared to packaged chips, having the bare die examined on a probe station resulted in comparatively minor temperature changes: the high thermal conductivity and large area of the silicon wafer and the stainless-steel electronic chuck where the bare chip was mounted, together with the relatively low on-chip power consumption and lack of packaging surrounding the chip, contributed to a highly efficient heat sink of the chip.

## Discussions

In summary, we demonstrated that directly injecting photocurrent pulses into multiple driver-less clock grids while monitoring timing with TDC can distribute electronic clocks in a CMOS chip with remarkable on-chip jitter and skew in the tens of femtoseconds regime. The elimination of clock drivers can significantly reduce on-chip heat dissipation from clock distribution as well.

For higher clock rates, we believe that microresonator-based Kerr combs[34] can be a viable option, ranging from few GHz[35] to tens GHz. We also recently found that the timing jitter of a 22 GHz silica Kerr micro-combs can be in the few-fs regime[36], which shows the potential of using such micro-combs for high-speed, low-jitter clocking applications. Another nice feature of such micro-combs is that they can be made on an integrated photonic platform, and eventually reduce the total size and cost.

While we showed the distribution of electronic clock signals using an injected photocurrent pulses from an off-chip photodiode in this work, on-chip photodiodes may be also used for realizing the same driverless on-chip clock distribution. It is worth noting that the supply voltage and the required clock loads of CMOS chips will decrease with the technology node and operation speed, resulting in the reduction of the required charge to drive the clocking part and making it suitable for driving by on-chip photodiodes. Recently, the co-integration of silicon photonics with CMOS functionalities has rapidly advanced: for example, recent work demonstrated the co-integration of 45 nm CMOS with 300 mm monolithic silicon photonic technology[37]. There have been also significant advances in on-chip photodiodes, particularly balanced on-chip photodiodes. For example, in ref. 38, they showed that a Ge on-chip BPD can generate 7 mA photocurrent (from each photodiode in the BPD) with bandwidth of 21 GHz. When applying 1 GHz (10 GHz) pulse train, it can generate 7 pC (0.7 pC) photocurrent pulse charge, which is sufficient to distribute clock signals for high-speed mixed-signal circuit blocks, such as data converters (ADCs and TDCs) and transceivers, via driverless metal structures. Even when using on-chip photodiodes, since the skew reduction is originated from the driverless property of the proposed CDN, it would not influence the performance. Regarding heat dissipation, the heat dissipation from on-chip photodiodes will not be separated from that of the electronics, and it will contribute to the overall heat dissipation of the CMOS chip. However, the clock load is generally small and at the same time, the power handling of the on-chip photodiode is limited to a few mW average power, limiting total heat dissipation to a few mW. Note that, considering that the H-tree in our small-scale test chip already dissipated ~22 mW of power, we believe that the heat dissipation from on-chip photodiodes will be still lower than the electronic counterpart.

On the other hand, the use of off-chip photodiodes, as shown in this work, still has the advantage of being able to drive a standard CMOS process-based electronics chip without requiring a special hybrid integration process. While femtosecond timing is still possible, this method is especially advantageous when on-chip heat dissipation is a critical problem. For lower-speed and larger clock load case, advanced off-chip photodiodes may be used to drive fairly large clock loads. For example, recent developments in balanced uni-traveling-carrier photodiodes enable the generation of even >100 mA average photocurrent output[39,40].

While the injection of photocurrent pulses from the off-chip photodiodes was realized by using microwave probes to obtain the main data of this work, chip-on-board (CoB), wire bonding, or flip-chip may be feasible solutions in practical applications. When the clock speed is low (e.g., < 1 GHz), the CoB might be the simplest approach. To assess feasibility, we also tested the injection of photocurrent pulses by using a standard CoB (see Methods and Supplementary Fig. 7). More advanced high-frequency-handling RF and microwave circuit board design[41] can be used to handle higher frequency signal. Wire-bonding of bare chips (i.e., chip-to-chip bonding between CMOS chip and photodiode chips)[42] may be also feasible.

The demonstrated approach can be applied in a variety of customizable ways depending on the intended applications and available fabrication processes. Timing 3D-stacked integrated circuits (3D-ICs) is one potential application. In the case of high bandwidth memory (HBM) devices[43] as an example, there are multiple layers of core memory dies with high-speed input/output (I/O) interfaces. Multi-layer core dies can benefit from low skew and low heat dissipation by injecting lower repetition-rate but higher pulse-energy optical pulse trains, while the high-speed I/O interfaces can benefit from higher repetition-rate optical pulse trains with ultralow timing jitter. In the case of advanced DRAM memory devices[44] where the typical clock speed is in the hundreds MHz range, the directly drivable clock load from one balanced uni-traveling-carrier photodiodes[39,40] can reach hundreds pF range. We believe that our method may be applied in a scalable way to drive larger clock loads by sectionizing the clock grids into multiple sections and driving each of them with a balanced photodiode (as demonstrated in this work) or by co-using clock drivers for small local distributions. For the high-speed I/O interface, where the required I/O clock speed is several GHz, on-chip BPDs or wire-bonded off-chip BPDs may be used to drive I/O circuit blocks.

Finally, we would like to briefly discuss on the overall electric power consumption and efficiency issue. As an example, for a 2.5 GHz repetition-rate laser, which is similar to the 2 GHz laser used in our work, it was recently reported that ~58 mW of output optical power was obtained with ~2 W electric power applied to the pump diode plus ~1 W electric power for thermoelectric cooling[45]. The total extractable photocurrent pulse charge (~10 pC) from the given optical power is capable of driving multiple transceivers or data converters. While our method may not be more energy efficient than electronic clock generation methods (for example, ~1.2 W power consumption for the case of AD9525 chip[46]), it is not considerably worse either, and it also offers unique benefits such as significantly lower jitter, skew, and heat dissipation in the chip. Furthermore, by utilising multiple fibre link branches and EDFAs (or Er-doped waveguide amplifiers (EDWAs)[47]), our method may be better suited for clocking many chips in data center-like environments. With recent advances in chip-scale microcombs and EDWAs, the microcomb-plus-many-EDWAs may be a compact and power-efficient option for distributing and injecting optical pulse signals to multiple chips in the near future.

## Methods
### Chip design
A test chip was fabricated in a 65 nm CMOS technology. The chip included three driver-less CDNs for the proof-of-principle experiment, a sub-ps time resolution stochastic TDC for skew measurement, D-flipflop-based clock loads, and a conventional H-tree CDN for comparison. The proposed driver-less CDNs have metal mesh structure implemented in the top metal and other circuit components were implemented under the CDNs. The three metal meshes (C0, C1, and C2) are 752 μm × 752 μm, 752 μm × 752 μm, and 1522 μm × 752 μm in size, respectively, and use M9 metal wire with 4 μm width and 22 μm mesh spacing. Clock outputs from the proposed driver-less CDN outputs are fed out through one clock driver (i.e., inverter) followed by an open-drain driver for jitter measurements. The H-tree CDN clocks are fed out

through two stages of clock drivers followed by an open drain driver as limited driving strength of the clock buffers require more stages to drive large open-drain driver. The H-tree CDN is placed under C2 to compare power, skew and jitter performances of the conventional H-tree with the proposed photonic CDNs. To distribute the clock signal across same area, the H-tree CDN required 23 stages of clock drivers (1194 in total). The D-flipflop-based load circuit consists of three D-flipflops connected in a loop so that the output switches every clock cycle (see Supplementary Fig. 4a). The test-chip prototype places 6036, 3360, and 3360 D-flipflops under C1, C2 and H-tree CDNs, respectively, to emulate the impact of active load variation due to transistor switching. The expected R and C parasitic values were extracted using Calibre-XRC (Siemens EDA)[28]. The driver-less CDNs also employ clamper circuits to avoid overdriving the capacitive load with excessive charges which can cause unbearably high on-chip voltages.

To measure the sub-ps time skew of photonic CDNs, we employed an 8-bit dual-supply stochastic TDC with ~100-fs time resolution[25] (Supplementary Fig. 4b). The TDC consists of 256 arbiters that can select between high/low supply voltages. As time offset in an arbiter varies with supply voltage, the TDC can effectively utilize 512 time-offsets for time-to-digital conversion, which enables sub-ps resolution. A scan-chain is employed to provide 1-bit supply selection signals (sel_vddh in Supplementary Fig. 4) to 256 arbiters and read out the raw 256-bit thermometer-coded TDC output at a slow rate. To enable both intra- and inter-domain CDN skew measurements, each one-bit arbiter employs a 5:1 multiplexer (Supplementary Fig. 4c). The multiplexer selects an input (i.e., start) out of five clocks from driver-less and H-tree CDNs to be compared with the reference (i.e., stop) NS0 signal. Prior to skew measurements, the TDC generates characteristic curves of each input by sweeping the delay between the input and reference (i.e., node NS0). The characteristic curves are used to map the skew measurement TDC output back to the time domain. As skew values of the proposed driver-less CDNs and conventional H-tree CDN have order of magnitude difference, TDC resolution was adjusted by changing supply voltages. For driver-less CDN (i.e., C1 and C2) skew measurements, nominal supply voltage setting (VDDH = 1.2 V, VDDL = 1.0 V in Supplementary Fig. 4c) is used that enabled 100-fs time resolution. For larger H-tree skew measurements, lower supply voltage setting (VDDH = 1.0 V, VDDL = 0.8 V) is used to allow broader input range with coarser (i.e., 600 fs) time resolution in order to deal with higher skew level.

### Chip test setup
The layout of the optical setup for chip test is shown in Supplementary Fig. 5. A 250 MHz mode-locked erbium-doped fibre oscillator is used as an optical frequency comb source. To multiply the repetition rate of the optical pulse signal to 500 MHz and 1 GHz, a two-stage Mach-Zehnder interferometer-based pulse repetition-rate multiplier (PRRM) is implemented. The resulting side-mode suppression ratio (SMSR) is 60 dB (Supplementary Fig. 6), which corresponds to 0.04% amplitude modulation depth. A dispersion compensating fibre (DCF) is used to reduce the pulsewidth of optical pulses incident on the photodiode down to ~250 fs. An erbium-doped fibre amplifier (EDFA) is placed after the PRRM to compensate for the losses by the PRRM. A 2 GHz Er-waveguide laser is used to generate 2 GHz clock. Then the optical pulses are split into two streams by an optical coupler. One steam is split once again and provides optical pulses for C0 and C1 clock domains; the other stream provides optical pulses for C2 clock domain. Motorized optical delay lines (MDLs) are inserted in the optical path to C1 and C2 clock domain to calibrate the TDC for skew measurement and active skew control. Programmable VOAs are inserted to adjust incident optical power for each photodiode. For photocurrent generation, 5 GHz BPDs (BDX1BA, Thorlabs) are used with ± 7 V bias. For 250 MHz clock generation, a single BPD is utilized to drive the clock load for all clock domains. For driving the full-swing clock voltage of 1 V, the optical power of ~4 mW is applied to BPDs for

C0 and C1, whereas ~8 mW is applied for C2 because of its larger clock load. Since the charge per pulse is inversely proportional to the pulse repetition rate at the same optical power, higher optical power is necessary for a higher repetition rate. As a result, ~8 mW is applied for C0 and C1 for 500 MHz clock generation. Because the maximum input power rating for the used BPD is 10 mW, two BPDs are used in parallel to drive C2 at 500 MHz. For 1 GHz clock generation, two BPDs are used for C0 and C1 and two or three BPDs are used for C2 (Supplementary Fig. 5). For 2 GHz clock generation, two BPDs with 10 mW incident power are used for C0. We experimentally confirmed that the chip can also operate well at lower optical power (for example, using two BPDs for C2 at 1 GHz and C0 at 2 GHz) with reduced clock voltage swing without degradation in jitter or skew performance. High-bandwidth RF probes are used to inject the photocurrent pulses toward the chip (Cascade Infinity quad, 100 μm pitch) and to take the output clock signal for off-chip jitter measurement (Unity, 100 μm pitch). The logic signals for skew measurement and supply voltages are also applied by the same probes (Supplementary Fig. 3).

## Waveform and jitter measurement

A clock driver followed by an open-drain driver is used for each clock domain for off-chip clock waveform and jitter measurement. The output from the chip is directly connected through RF cables to a 33 GHz bandwidth, 128 GSa/s real-time oscilloscope (UXR0334A, Keysight). For the trigger signal, the photocurrent pulse signal extracted from a 12 GHz p-i-n photodiode is used. A trigger signal is divided by a power splitter to quantify the jitter histogram measurement limit of the utilized oscilloscope; one is used as a trigger signal and the other is used to measure the instrumental limit as shown in Fig. 3b. For jitter measurement consistency, half of the photocurrent pulse signal is utilized as the trigger signal, with the other power splitter output terminated by a 50 ohm resistor. To determine the jitter histogram's standard deviation and peak-to-peak value, 100,000-hits timing instants when the signal crosses at half its highest amplitude are collected. Note that there was no observable difference in jitter level even when the dynamic clock load is applied (load on/off in Supplementary Fig. 2).

## Photocurrent pulse jitter measurement using EOS-TD

To measure the timing jitter of photocurrent pulses used to generate a digital clock, an electro-optic sampling-based timing detector (EOS-TD)[48,49] is used. A fibre delay placed in front of EOS-TD changes the relative temporal position between the optical and photocurrent pulses. The EOS-TD output is measured by a fast Fourier transform (FFT) analyser (Stanford Research Systems, SR770) and a radio-frequency (RF) spectrum analyser (Agilent, E4411B) for 100 Hz–100 kHz and 100 kHz–10 MHz offset frequency ranges, respectively. The integrated timing jitter is obtained by integrating the jitter PSD. To estimate the clock jitter transferred from the photocurrent pulse jitter, the timing jitter of the photocurrent pulse is evaluated at different temporal positions. The clock output is obtained by integrating the photocurrent pulse input (curve (i) of Fig. 4b). When that clock output reaches half of its maximum amplitude, the photocurrent pulse crosses 90% point of its maximum amplitude in the falling-edge, which is denoted as B in the curve (i) of Fig. 4b.

## Skew measurement and adaptive skew control

On-chip TDC measures inter-domain skew between C0 and other domains (C1, C2 and H-tree). Specific node locations where the TDC takes the arriving clock signals are denoted in Fig. 5a. A pattern generator (PG, PkPG2116 + , Acute) generates the required logic signals to enable skew measurement and sends them through the scan chain. A local analyser (LA, PKLA1116, Acute) receives the output of TDC through the same scan chain (Supplementary Fig. 3). The PG and LA are operated at 10 MS/s and 20 MS/s, respectively. Dynamic clock loads are turned on for the entire skew measurements. For TDC calibration, a

time offset is provided to one of the clock signals by a motorized delay line (MDL) inserted in one optical signal path, which generates a certain amount of static skew, and the TDC output is monitored at the same time. The TDC outputs are measured with an interval of 100 fs and this process is repeated for 10 times, yielding characteristic curves. For intra-domain skew measurement (Fig. 5b), data points at the characteristic curves of the left and right clock nodes (e.g., NS2L and NS2R for C2 in Fig. 5a) are measured one by one alternately through the entire range. The time offset is then set to the average of midpoints of each characteristic curve. For a total of 1000 skew data, the TDC output is collected one by one for each clock node alternately, which takes around 160 min per node. By subtracting the inter-domain skew data from both nodes, the intra-domain skew can be obtained. The inter-domain skew shown in Fig. 5c is measured at time offset when the output code corresponds to the midpoint of a characteristic curve. The inter-domain skew can be compensated in the optical domain by the MDL. The TDC is used to assess inter-domain skew, and a feedback signal of roughly 0.1 Hz data-rate is provided to the MDL for adaptive skew control (Fig. 5c). Note that this low rate is mainly dominated by communication time between LA, PG and used PC for data acquisition.

## Power consumption calculation and thermal imaging

An infrared camera (A35, FLIR) with temperature resolution of 0.05 °C is used to measure the surface temperature of the chip to evaluate the thermal loading of the CDN. The horizontal and vertical fields of vision were 40 cm and 30 cm, respectively, corresponding to a 0.11 mm spatial resolution. Each measuring environment was retained for an hour before temperature measurements to achieve thermal equilibrium.

## Chip-on-board implementation and testing

We designed a printed circuit board (PCB) to demonstrate the photocurrent injection with a CoB method. After wire-bonding pads of the bare chip to the PCB, the BPD outputs were connected to the chip by the PCB trace (see Supplementary Fig. 7a). Note that, in this demonstration, the package of the used commercial BPD was much larger (12 mm by 10 mm footprint) than the test chip (2 mm by 2 mm footprint) and because we designed the chip with pin arrangements optimized for RF probes for testing purposes, it was not possible to make the traces to the chip shorter than ~4 mm, limiting the achievable clock speed. The measured clock waveforms and jitter data for 250 MHz and 1 GHz clock speed examples are shown in Supplementary Fig. 7.

## Data availability

The authors declare that the data supporting the findings of this study are available within the paper, supplementary information and source data files.

## Code availability

The simulation and computational codes for this study are available from the corresponding authors on request.

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

## Acknowledgements
We thank Prof. Joo-Young Kim and Prof. Kyoungsik Yu at KAIST for fruitful discussions. This research was supported by the Samsung Research Funding and Incubation Center for Future Technology (Grant no. SRFC-IT1702-53 for J.K. and H.C.).

## Author contributions
H.C. and J.K. conceived the idea and supervised the project. H.C. designed the CMOS chip. M.H. designed and implemented the optical test setup. H.C. and W.N. ran simulations for chip verification. M.H. and H.C. tested the chip and obtained data. All authors analysed the data. M.H., H.C., and J.K. wrote the manuscript with inputs from all authors.

## Competing interests
H.C. and J.K. are inventors on a patent related to this work filed by Korea Advanced Institute of Science and Technology (KAIST) and Korea University Sejong Campus (Korean patent 10-2161837 issued on September 24, 2020). The authors declare no other competing interests.
