## [Peer Review File · Nature Communications]

REVIEWER COMMENTS

Reviewer #1 (Remarks to the Author):

The authors present an optical clock distribution network (CDN) on-chip that provides synchronized clock signals to different circuit blocks on the chip. Today's on-chip CDNs distribute electronic signals, resulting in increased jitter, skew, and heat dissipation due to the necessary clock drivers. In this work low-jitter optical pulses are locally injected into the chip for an effective distribution of high-quality clock signals. This has never been studied before. The authors succeed to demonstrate femtosecond-precision distribution of electronic clock signals using driver-less CDNs combining combining ultralow comb-jitter, driver-less metal-meshes, and active skew control. This work shows the potential of photonic clock distribution inside high-performance integrated circuits, which is crucial for many electronic and the upcoming photonic-electronic chips. An added value is also the much lower heat dissipation on chip because of the removal of electronic clock drivers in a fully optical power supplied system.

However, a few questions remain that should be clarified before publication.

1. The authors write very often, that an optical frequency comb signal is injected in the chip. Is it necessary that the pulse trains come from an optical frequency comb? I would guess one only needs a low jitter pulse train but not a frequency comb. Please clarify.
2. In this work no optical pulses are injected but rather the differential output from balanced receivers are injected into the integrated circuit. What is the thought of the authors for real world applications for an on chip CDN. Would in an actual chip the photodetectors also be integrated and the optical signals be distributed on chip via optical waveguides?
3. Is the optical power high enough to completely replace clock drivers, such that driver jitter can be avoided in actual circuits that need very low jitter like analog-to-digital convertors? Or this only true for this test chip. So the question is really whether on-chip driverless supply of advanced electronic chips is possible in reality by direct optical drive power via photo-detection? Please provide an example with a rough power budget to show that this can be applied to real world electronic systems.

If those questions can be satisfactorily answered, I think this is a very high impact paper and should be published.

Reviewer #2 (Remarks to the Author):

Review of “Femtosecond-precision electronic clock distribution in CMOS chips by injecting optical frequency combs” by Hyun et al.

This paper demonstrates use of an optical frequency comb (mode locked laser) as a low-jitter optical clock for a CMOS chip, the goal being to reduce the power consumption, timing jitter and skew (spatial inhomogeneity of arrival times) in the chip clock distribution network (CDN). The authors design a 65nm CMOS chip with electrical clock distribution trees but having no drivers, instead directly driven from photodiodes illuminated with arriving laser pulses. They also have a baseline case with electronic drivers for comparison. They also show adaptive skew control, by detecting arrival time differences on chip and providing feedback to an off-chip optical delay line. Last, they use a thermal camera to show a temperature difference with photonic vs. on-chip electrical CDN drive due to differing power consumption.

I find that the paper presents a high level of scientific rigor and quality, is well written, and the presented technical conclusions are well supported by the presented data and figures. The timing jitter results are excellent showing 300fs order jitter (limited by measurement) vs. ~2.5ps jitter in the baseline electronic example. The adaptive skew control is interesting, but it is unclear if such a scheme is practical given that long tunable delays are difficult to implement in integrated photonics (presumably the eventual implementation of such control). The temperature demonstration is not very significant, as temperature rise is <1-2 deg C in both cases – as the authors themselves note.

This subject area has a considerably history despite papers on it being sparse i.e. few and far between. This work represents an advance in that an optical frequency comb is driving a CMOS chip. Previously both mode locked lasers were considered, and lasers were used to drive a chip (as the authors' referenced papers show – Ref 19-23 ish), but a low jitter source was not used to demonstrate the reduction in jitter.

I think this work deserves publication in a journal visible to the community. Whether its significance deserves the wide distribution offered by publication in Nature Communications I am on the fence about.

I have the following questions for the authors:

1. Abstract: “effective distribution of such high-quality clock signals has not been actively studied yet”. Statement is a bit strong given the literature cited.

2. Authors cite CDN clock drivers power consumption “accounting for 10-30% of overall chip power consumption and causing on-chip thermal problems”. However the power and thermal impact demonstrated in the paper is small. The authors mention the absent die packaging one factor. Some comment by the authors on the disconnect between the demonstrated differences and this claim would be helpful.

3. The photodiodes are off chip, and the signals are delivered by GSG microwave probes. How do the authors envision their demonstration translating to actual applications? Will you need RF cables going to the chip package? Will you use wirebonds to another photonic chip – in which case, will such wirebonds not produce issues in the pulse dispersion and jitter? Or, will they need to be integrated – however, the used photodiodes were already power limited at 5mW and had to be parallelized for the higher rep rate demos – it is not clear that on-chip photodiodes are capable of such power handling (and, on chip photodiodes would require a monolithic electronics-photonics platform, different from the process used in this paper).

4. The authors claim that the presented approach can “significantly reduce the skew and on-chip heat dissipation”. Would this still be true if the pulses (which have considerable power) are detected in on-chip photodiodes? And, if on-chip photodiodes are not the intended approach, then how does the scheme hold together with off-chip ones.

5. Re: power consumption, the on-chip CDN power is reduced, but the additional cost is powering an off-chip optical frequency comb. How does this power comparison work, and is it a realistic future tradeoff? For example the comb is 5mW average power, it is probably 0.1 to 1% wall plug efficiency or something like that, so that’s about 0.5W – 5W.. is that right? What are the prospects of eventual optical clock distribution? A paragraph addressing this would be helpful (which really covers several points above).

6. The authors use external delay lines to multiply up the frequency comb rep rate. Is this a viable approach for eventual integration of a high rate clock source?

My general feeling is that the demonstration is effective, but that the relevance of this approach to actual implementation of optical clock distribution still leaves a number of questions open, that would benefit from the authors’ comment in the manuscript, and that define the context in which the significance of the current work can be better evaluated.

Point-by-point response to reviewers' comments

Reviewer 1

The authors present an optical clock distribution network (CDN) on-chip that provides synchronized clock signals to different circuit blocks on the chip. Today's on-chip CDNs distribute electronic signals, resulting in increased jitter, skew, and heat dissipation due to the necessary clock drivers. In this work low-jitter optical pulses are locally injected into the chip for an effective distribution of high-quality clock signals. This has never been studied before. The authors succeed to demonstrate femtosecond-precision distribution of electronic clock signals using driver-less CDNs combining combining ultralow comb-jitter, driver-less metal-meshes, and active skew control. This work shows the potential of photonic clock distribution inside high-performance integrated circuits, which is crucial for many electronic and the upcoming photonic-electronic chips. An added value is also the much lower heat dissipation on chip because of the removal of electronic clock drivers in a fully optical power supplied system. However, a few questions remain that should be clarified before publication.

We thank the reviewer has taken the time to review the submitted manuscript and provided constructive suggestions and comments on our manuscript.

Point 1-1

The authors write very often, that an optical frequency comb signal is injected in the chip. Is it necessary that the pulse trains come from an optical frequency comb? I would guess one only needs a low jitter pulse train but not a frequency comb. Please clarify.

As the reviewer correctly pointed out, the demonstrated method requires a low-jitter optical pulse train in the time domain but not necessarily the frequency-domain property of a frequency comb structure. The reason why we used the term “frequency comb” in the original manuscript was to emphasize the type of sources capable of generating such low-jitter sub-picosecond optical pulses. Please note that, while we used femtosecond passively mode-locked lasers (a type of frequency comb) in this work, other (usually referred to) *comb* sources, such as Kerr combs and electro-optic combs (when driven by low-jitter microwaves), can also be used to drive the chip. We intended to refer to these sources collectively as frequency combs.

Revisions made:

We replaced “optical frequency comb” to “optical pulse train” or “photocurrent pulses” in the title and other places throughout the manuscript where it is more appropriate. We limited the use of “optical frequency comb” only where it really means the general comb sources that can generate low-jitter optical pulse trains.

Point 1-2

In this work no optical pulses are injected but rather the differential output from balanced receivers are injected into the integrated circuit. What is the thought of the authors for real world applications for an on chip CDN. Would in an actual chip the photodetectors also be integrated and the optical signals be distributed on chip via optical waveguides?

Indeed, in this work, we showed the distribution of electronic clock signals using an injected photocurrent pulses from an off-chip photodiode. As the reviewer commented, on-chip photodiodes may be also used, and we believe that there can be many different ways to use the demonstrated driver-

less clock distribution idea, depending on the intended applications. Here, let us elaborate more on the potential application scenarios.

First, we can employ the off-chip photodiodes to drive standard CMOS chips, as demonstrated in this work. This approach has the advantage of being able to drive a standard CMOS process-based electronics chip without requiring a special hybrid integration process. While femtosecond timing is still possible, this method is especially advantageous when on-chip heat dissipation is a critical problem. For example, the memory core of high-bandwidth memory (HBM) DRAM devices, which are relatively low speed (~200-500 MHz clock speed) and can have a heat issue due to the 3D-stacked structure, can benefit greatly. Multiple high-power off-chip photodiodes can be used to drive clock loads with significantly reduced heat dissipation while maintaining good timing (which will be also discussed below in the response to Point 1-3).

Second, as mentioned by the reviewer, the use of optical clocking in an electronic-photon hybrid chip can be also useful, particularly for high-speed applications such as high-speed data converters and I/O interfaces. In this case, on-chip photodiodes (and, if necessary, waveguides) can be integrated on the CMOS chip. Recently, the co-integration of silicon photonics with CMOS functionalities has rapidly advanced. For example, recent work demonstrated the co-integration of 45-nm CMOS with 300-nm monolithic silicon photonic technology (Rakowski et al, *OFC 2020*, Paper T3H.3), demonstrating a significant potential in this direction. In this case, the driver-less metal structure can be still used to distribute electronic clock signals generated directly by the on-chip photodiodes. As the high-speed circuits have a relatively small clock load, they can be driven by either on-chip or off-chip photodiodes (we will discuss more on this power and load issue in the response to Point 1-3).

Based on the demonstration results of our work and the reasoning outlined above, we believe that both on-chip and off-chip photodiode approaches, either separately or combined ways, can be effective for distributing high-quality clock signals in the CMOS chips with significantly low on-chip heat dissipation. It can be used in many different, flexible ways depending on the intended applications and available fabrication processes.

Revisions made: As the revision is closely connected with Point 1-3, we combine and summarize the revisions that we have made at the end of response to Point 1-3.

Point 1-3

Is the optical power high enough to completely replace clock drivers, such that driver jitter can be avoided in actual circuits that need very low jitter like analog-to-digital convertors? Or this only true for this test chip. So the question is really whether on-chip driverless supply of advanced electronic chips is possible in reality by direct optical drive power via photo-detection? Please provide an example with a rough power budget to show that this can be applied to real world electronic system.

We thank the reviewer to give the opportunity to consider this important aspect. Thanks to the EDFAs and EDWAs (for example, Liu et al, *Science* **376**, 1309 (2022)), optical power supply can be sufficient, which is the same case for the present demonstration. Thus, the drivable on-chip load is determined by the power handling capability of the photodiodes. As discussed in the response to Point 1-2, we can consider two major cases in terms of optical power and drivable clock load: (a) low-speed and large clock load case (where on-chip heat issue matters more) and (b) high-speed and small clock load case (where timing matters more).

For the low-speed and large load case, we believe that advanced off-chip photodiodes can be used to drive fairly large clock loads with minimal on-chip heat dissipation and skew by eliminating or minimizing

the number of on-chip clock drivers. For example, recent developments in balanced uni-traveling-carrier photodiodes enable the generation of even >100 mA average photocurrent output (Li et al, IEEE Photon. Tech. Lett. **21**, 1858 (2009); Houtsma et al, *ECOC 2011*, Paper Tu.3.LeSaleve.6). Because the amount of charge per photocurrent pulse determines the drivable clock load by $C_L=Q/V_{dd}$, in the case of advanced DRAM memory devices where the typical clock speed is in the 200 MHz – 500 MHz range, the directly drivable clock load from one balanced photodiode can reach hundreds pF range (e.g., 200 pF when $f = 500$ MHz, $V_{dd} = 1$ V, and $i_{avg} = 100$ mA). We believe that our method may be applied in a scalable way to drive larger clock loads with minimal on-chip heat dissipation while maintaining fs-level timing by sectionizing the clock grids into multiple sections and driving each of them with a balanced photodiode (as demonstrated in this work) or use clock drivers only for small local distributions.

For the high-speed and low-jitter case, it is worth noting that the supply voltage (V_{dd}) and the required clock loads of CMOS chips will decrease with the technology node and operation speed, which results in the reduction of the required charge to drive the clocking part. Thus, they may be driven by either an on-chip photodiode or an off-chip photodiode. Recently, there have been significant advances in on-chip photodiodes, particularly balanced on-chip photodiodes. For example, in Tzu et al, IEEE JSTQE **25**, 3800111 (2019) paper, they showed that a Ge on-chip BPD can generate 7-mA photocurrent (from each photodiode in the BPD) with bandwidth of 21 GHz. When applying 1-GHz (10-GHz) pulse train, it can drive up to 7 pF (0.7 pF) clock load (when $V_{dd} = 1$ V). For the example of advanced DRAMs again, where the required I/O clock speed is several GHz, a single BPD may be sufficient to drive I/O circuit block. If necessary, multiple on-chip photodiodes can be also used for driving multiple clock load sections. With a short (<1 mm) and well-designed wire-bonding between the CMOS chip and the off-chip photodiode (for example, as shown in Li et al, Opt. Express **28**, 14038 (2020)), we believe that high-speed and high-power off-chip BPDs may be also used to generate high-speed (several to tens GHz) clock signals for high-speed circuit elements.

In fact, in our test chip, we already showed that the photocurrent pulse injection and driverless CDN approach could drive not only digital loads (D-flipflops) but also the sub-ps-resolution time-to-digital converter (TDC) circuit block itself. Therefore, we believe that our approach can be easily applied to distribute clock signals for high-speed mixed-signal circuit blocks such as data converters (ADCs and TDCs) and transceivers.

Revisions made: We added the following paragraphs in the Discussion section to address reviewer’s comments.

- A paragraph on the potential use of on-chip photodiodes (pages 9-10; underlined red fonts): “While we showed the distribution of electronic clock signals using an injected photocurrent pulses from an off-chip photodiode in this work, on-chip photodiodes may be also used for realizing the same driverless on-chip clock distribution. It is worth noting that the supply voltage and the required clock loads of CMOS chips will decrease with the technology node and operation speed, resulting in the reduction of the required charge to drive the clocking part and making it suitable for driving by on-chip photodiodes. Recently, the co-integration of silicon photonics with CMOS functionalities has rapidly advanced: for example, recent work demonstrated the co-integration of 45-nm CMOS with 300-nm monolithic silicon photonic technology³⁷. There have been also significant advances in on-chip photodiodes, particularly balanced on-chip photodiodes. For example, in ref. 38, they showed that a Ge on-chip BPD can generate 7-mA photocurrent (from each photodiode in the BPD) with bandwidth of 21 GHz. When applying 1-GHz (10-GHz) pulse train, it can generate 7 pC (0.7 pC) photocurrent pulse charge, which is sufficient to distribute clock signals for high-speed mixed-signal circuit blocks, such as data converters (ADCs and TDCs) and transceivers, via driverless metal structures.”

- A paragraph on the use of high-power off-chip photodiodes (in page 10; underlined red fonts): “On the other hand, the use of off-chip photodiodes, as shown in this work, still has the advantage of being able to drive a standard CMOS process-based electronics chip without requiring a special hybrid integration process. While femtosecond timing is still possible, this method is especially advantageous when on-chip heat dissipation is a critical problem. For lower-speed and larger clock load case, advanced off-chip photodiodes may be used to drive fairly large clock loads. For example, recent developments in balanced uni-traveling-carrier photodiodes enable the generation of even >100 mA average photocurrent output^{39,40}.”
- More detailed discussion of the potential application scenario of both on-chip and off-chip photodiodes (in page 11; underlined red fonts): “In the case of advanced DRAM memory devices⁴⁴ where the typical clock speed is in the hundreds MHz range, the directly drivable clock load from one balanced uni-traveling-carrier photodiodes^{39,40} can reach hundreds pF range. We believe that our method may be applied in a scalable way to drive larger clock loads by sectionizing the clock grids into multiple sections and driving each of them with a balanced photodiode (as demonstrated in this work) or by co-using clock drivers for small local distributions. For the high-speed I/O interface, where the required I/O clock speed is several GHz, on-chip BPDs or wire-bonded off-chip BPDs may be used to drive I/O circuit blocks.”

Point 1-4

If those questions can be satisfactorily answered, I think this is a very high impact paper and should be published.

We would like thank Reviewer 1 once again for recognizing the significance and high technical quality of our work, and hope that the referee finds our responses to be satisfactory.

Reviewer 2

This paper demonstrates use of an optical frequency comb (mode locked laser) as a low-jitter optical clock for a CMOS chip, the goal being to reduce the power consumption, timing jitter and skew (spatial inhomogeneity of arrival times) in the chip clock distribution network (CDN). The authors design a 65nm CMOS chip with electrical clock distribution trees but having no drivers, instead directly driven from photodiodes illuminated with arriving laser pulses. They also have a baseline case with electronic drivers for comparison. They also show adaptive skew control, by detecting arrival time differences on chip and providing feedback to an off-chip optical delay line. Last, they use a thermal camera to show a temperature difference with photonic vs. on-chip electrical CDN drive due to differing power consumption.

I find that the paper presents a high level of scientific rigor and quality, is well written, and the presented technical conclusions are well supported by the presented data and figures. The timing jitter results are excellent showing 300fs order jitter (limited by measurement) vs. ~2.5ps jitter in the baseline electronic example. The adaptive skew control is interesting, but it is unclear if such a scheme is practical given that long tunable delays are difficult to implement in integrated photonics (presumably the eventual implementation of such control). The temperature demonstration is not very significant, as temperature rise is <1-2 deg C in both cases – as the authors themselves note.

This subject area has a considerably history despite papers on it being sparse i.e. few and far between. This work represents an advance in that an optical frequency comb is driving a CMOS chip. Previously both mode locked lasers were considered, and lasers were used to drive a chip (as the authors' referenced papers show – Ref 19-23 ish), but a low jitter source was not used to demonstrate the reduction in jitter.

I think this work deserves publication in a journal visible to the community. Whether its significance deserves the wide distribution offered by publication in Nature Communications I am on the fence about.

We thank the reviewer for taking the time to review our manuscript thoroughly. We hope that our revision could address the reviewers' concerns.

Point 2-1

Abstract: "effective distribution of such high-quality clock signals has not been actively studied yet". Statement is a bit strong given the literature cited.

We agree, and following the reviewer's suggestion, we changed the abstract in the following way.

Revisions made: We changed the abstract as "While low-jitter optical pulses have been locally injected in the chip, research on effective distribution of such high-quality clock signals has been relatively sparse."

Point 2-2

Authors cite CDN clock drivers power consumption "accounting for 10-30% of overall chip power consumption and causing on-chip thermal problems". However, the power and thermal impact demonstrated in the paper is small. The authors mention the absent die packaging one factor. Some comment by the authors on the disconnect between the demonstrated differences and this claim would be helpful.

Typical power consumption for CDNs is known to be in the 10-30% range for processors and digital systems, as indicated by refs. 6-9 and many other literatures. For more general circuits and chips, the on-chip clocking power varies widely and it is difficult to generalize the percentage of power consumption in clocking, although it will certainly require a significant amount of on-chip power consumption.

Our chip size was relatively small in comparison to large-scale microprocessor chips, and as a result, the chip power consumption and thermal impact were also relatively small. Nonetheless, by measuring the power supply voltage and current, we were able to measure the overall power consumption and power consumed in H-tree CDN operation in the test chip. For the H-tree (which shares the same area with C2 on the test chip), the overall power consumption and CDN power consumption were 51.5 mW ($=1.2 \text{ V} \times 42.9 \text{ mA}$) and 22.1 mW ($=1.2 \text{ V} \times 18.42 \text{ mA}$), respectively. As a result, while the power consumption itself was small, the relative ratio of electronic CDN power consumption was significant ($\sim 43\%$). Note that, by using the extracted resistance of the metal grid of C2 section (0.16 ohm) and the measured photocurrent pulse waveform, we could estimate the on-chip power dissipation for optical distribution case, which resulted in $\sim 112 \mu\text{W}$ for 1-GHz clocking case.

In the demonstration experiment, the on-chip temperature change was not significant due to the relatively low on-chip power consumption and the usage of a bare die chip mounted on a silicon wafer positioned on a probe station. The high thermal conductivity and large area of the silicon wafer and stainless-steel electronic chuck, where the chip was mounted, together with the lack of packaging surrounding the chip, contributed to a highly efficient heat sink of the chip, and the temperature rise was not very significant. Despite these issues, as shown in Fig. 4, we could clearly observe a difference in heat dissipation and temperature rise between optical and electronic clock distribution methods.

Revisions made:

- In the “accounting for 10-30% of overall chip power consumption and causing on-chip thermal problems” sentence in page 2, we clarified that it is generally applicable for processors and digital systems.
- We expanded the section on on-chip power dissipation and thermal dissipation in page 8. We added the measured and calculated on-chip power consumption of the H-tree and photocurrent pulse injection cases (“We first assessed the overall power consumption and CDN power consumption when driven by the H-tree CDN. The measured overall power consumption and CDN power consumption were 51.5 mW and 22.1 mW, respectively, which shows that the relative ratio of CDN power consumption was $\sim 43\%$ for the test chip. In contrast, the on-chip power dissipation for the driver-less distribution in C2 was calculated as $\sim 112 \mu\text{W}$ for 1-GHz clocking, which is significantly lower than the H-tree case, by using the extracted resistance of the metal grid of C2 (0.16 ohm) and the measured current pulse waveforms”).
- We also added more discussion on the reasons on a relatively small temperature change of the bare chip test (“the high thermal conductivity and large area of the silicon wafer and the stainless-steel electronic chuck where the bare chip was mounted, together with the relatively low on-chip power consumption and lack of packaging surrounding the chip, contributed to a highly efficient heat sink of the chip”).

Point 2-3

The photodiodes are off chip, and the signals are delivered by GSG microwave probes. How do the authors envision their demonstration translating to actual applications? Will you need RF cables going to the chip package? Will you use wirebonds to another photonic chip – in which case, will such

wirebonds not produce issues in the pulse dispersion and jitter? Or, will they need to be integrated – however, the used photodiodes were already power limited at 5mW and had to be parallelized for the higher rep rate demos – it is not clear that on-chip photodiodes are capable of such power handling (and, on chip photodiodes would require a monolithic electronics-photonics platform, different from the process used in this paper).

As the reviewer rightly commented, the injection of photocurrent pulses from the off-chip photodiodes was realized by using microwave probes in our demonstration. As also discussed in the response to Point 1-3 of Reviewer 1, we believe that both on-chip and off-chip photodiodes can be used for driver-less CDN technology. Here let us discuss on the potential future directions that we envision for both off-chip and on-chip photodiode cases below.

For the off-chip photodiode case, chip-on-board (CoB), wire bonding, or flip chip may be feasible solutions in practical applications. When the clock speed is low (e.g., <1 GHz), the CoB might be the simplest approach. In this revision, we designed a printed circuit board (PCB) to demonstrate the photocurrent injection with a CoB method. After wire-bonding pads of the bare chip to the PCB, the BPD outputs were connected to the chip by the PCB trace (see the photo in Fig. L1 below). Note that, in this demonstration, the package of the used commercial BPD was much larger (12 mm by 10 mm footprint) than the test chip (2 mm by 2 mm footprint) and because we designed the chip with pin arrangements optimized for RF probes for testing purposes, it was not possible to make the traces to the chip shorter than ~4 mm, limiting the achievable clock speed.

Fig. L1. Photo of the CoB test board

The measured clock waveforms and jitter data for 250 MHz and 1 GHz clock speed examples are shown in Fig. L2. Compared to the RF probe cases, the clock waveforms are slightly worse with some oscillations, owing to reflection issues caused by the non-optimized PCB trace lengths between the CMOS chip and the BPD outputs. While the jitters of the 250-MHz cases are not significantly different (limited by the oscilloscope performance), the jitter of the 1-GHz CoB case is slightly higher than that of the RF probe case. We believe that the main reason is the reflection, and that jitter performance is degraded because higher frequency is influenced more, rather than the use of the PCB itself. If the PCB trace length can be optimized, we anticipate that the jitter performance to be comparable to that of the RF probe case.

Fig. L2. Clock waveforms and measured timing jitter for (a) 250 MHz and (b) 1 GHz clock rate

While the typical PCB was used and the PCB trace length was unoptimized due to the photodiode package issue in this revision, more advanced high-frequency-handling RF and microwave circuit board design [IPC-2252 Design Guide for RF / Microwave Circuit Boards], combined with smaller package photodiodes and optimized pin arrangement of the CMOS chip, can be used to handle higher frequency signal with optimal routing by a software (for example, Keysight ADS).

Wire-bonding of bare chips (i.e., chip-to-chip bonding between CMOS chip and photodiode chips) may be also feasible. As an example, shown in Li et al, *Opt. Express* **28**, 14038 (2020), the CMOS chip and the 30-GHz photodiode were connected by a $\sim 350\text{-}\mu\text{m}$ -long wire bonding that was optimized using Keysight ADS software. Photocurrent pulse injection with tens of GHz speed may be achieved with such short and optimized wire bonding in conjunction with microwave circuit board design. In this case, we also believe that the wire-bonding dispersion will not limit the clocking performance.

For higher-speed clock injection, on-chip photodiodes may be more advantageous. In general, the clock load for higher speed clocks is lower, and as mentioned in the response to Reviewer 1's Point 1-3, it may be possible to drive a fair amount of clock loads on the chip (up to several to tens pF load depending on the clock speed) to distribute clock signals for high-speed mixed-signal circuit blocks such as data converters (ADCs and TDCs) and transceivers.

We anticipate that the effective usage of both on-chip and off-chip photodiode approaches will eventually maximize the usefulness of our technology, necessitating additional future research. The significance of our work is that it demonstrated the effectiveness of a driver-less on-chip clock distribution strategy capable of distributing on-chip digital clocks with femtosecond jitter and skew while also significantly reducing on-chip heat dissipation.

Revisions made:

1. We added the CoB clock generation result (Figs. L1 and L2 in this letter) as Supplementary Figure 7 in the revised paper.
2. We added the CoB clock generation method in Methods section (in page 17; underlined red fonts): "**Chip-on-board implementation and testing.** We designed a printed circuit board (PCB) to demonstrate the photocurrent injection with a CoB method. After wire-bonding pads of the bare chip to the PCB, the BPD outputs were connected to the chip by the PCB trace (see Supplementary Fig. 7a). Note that, in this demonstration, the package of the used commercial BPD was much larger (12 mm by 10 mm footprint) than the test chip (2 mm by 2 mm footprint)

and because we designed the chip with pin arrangements optimized for RF probes for testing purposes, it was not possible to make the traces to the chip shorter than ~4 mm, limiting the achievable clock speed. The measured clock waveforms and jitter data for 250 MHz and 1 GHz clock speed examples are shown in Supplementary Fig. 7.”

3. We added discussion on the use of CoB and wire-bonding in Discussion section (in pages 10-11; underlined red fonts): “While the injection of photocurrent pulses from the off-chip photodiodes was realized by using microwave probes to obtain the main data of this work, chip-on-board (CoB), wire bonding, or flip-chip may be feasible solutions in practical applications. When the clock speed is low (e.g., <1 GHz), the CoB might be the simplest approach. To assess feasibility, we also tested the injection of photocurrent pulses by using a standard CoB (see Methods and Supplementary Fig. 7). More advanced high-frequency-handling RF and microwave circuit board design⁴¹ can be used to handle higher frequency signal. Wire-bonding of bare chips (i.e., chip-to-chip bonding between CMOS chip and photodiode chips)⁴² may be also feasible.”

Point 2-4

The authors claim that the presented approach can “significantly reduce the skew and on-chip heat dissipation”. Would this still be true if the pulses (which have considerable power) are detected in on-chip photodiodes? And, if on-chip photodiodes are not the intended approach, then how does the scheme hold together with off-chip ones.

First, regarding skew, since the skew reduction is originated from the driver-less property of the proposed CDN, we expect that injecting methods, whether using off-chip or on-chip photodiodes, would not influence the performance. The skew between different clock grid sections can be also monitored and compensated by tuning the relative optical delay to each section.

Regarding heat dissipation, we understand the reviewer’s concern about the heat dissipation of on-chip photodiodes: indeed, the heat dissipation from on-chip photodiodes will not be separated from that of the electronics, and it will contribute to the overall heat dissipation of the CMOS chip. As we also outlined in the response to Reviews 1-2 and 1-3, that is why we believe that, when the clock load is large and the heat dissipation is a main concern, it might be a better approach to use high power-handling off-chip photodiodes. For higher-speed and low-jitter applications, where on-chip photodiodes may be more suitable, the clock load is generally smaller and at the same time, the power handling of the on-chip photodiode is limited to a few mW average power, limiting total heat dissipation to a few mW at most. Note that, as answered in the response to Review 2-2, even in this small chip, considering that the H-tree dissipate ~22 mW of power, we believe that the heat dissipation will be lower than the electronic counterpart when using an on-chip photodiode.

Revisions made: We added discussion on this issue in Discussion section (in page 10; underlined red fonts): “Even when using on-chip photodiodes, since the skew reduction is originated from the driverless property of the proposed CDN, it would not influence the performance. Regarding heat dissipation, the heat dissipation from on-chip photodiodes will not be separated from that of the electronics, and it will contribute to the overall heat dissipation of the CMOS chip. However, the clock load is generally small and at the same time, the power handling of the on-chip photodiode is limited to a few mW average power, limiting total heat dissipation to a few mW. Note that, considering that the H-tree in our small-scale test chip already dissipated ~22 mW of power, we believe that the heat dissipation from on-chip photodiodes will be still lower than the electronic counterpart.”

Point 2-5

Re: power consumption, the on-chip CDN power is reduced, but the additional cost is powering an off-chip optical frequency comb. How does this power comparison work, and is it a realistic future tradeoff? For example the comb is 5mW average power, it is probably 0.1 to 1% wall plug efficiency or something like that, so that's about 0.5W – 5W.. is that right? What are the prospects of eventual optical clock distribution? A paragraph addressing this would be helpful (which really covers several points above).

We thank the reviewer to give the opportunity to consider this important aspect of our system. As the reviewer correctly noted, the wall-plug efficiency of mode-locked lasers or frequency combs in general is relatively low, ranging from $\sim 0.1\%$ to $\sim 10\%$. As an example, for a 2.5-GHz repetition-rate Er-gain medium-based laser [shown in Emaury, F. et al, *Proc. Frontiers in Ultrafast Optics: Biomedical, Scientific and Industrial Applications XXII*, PC119910P (2022, SPIE)], which is similar to the 2-GHz laser used in our work, ~ 58 mW of output optical power was obtained with ~ 2 W electric power (1000 mA current) applied to the pump diode plus ~ 1 W electric power for TEC cooling of the pump diode, corresponding to ~ 3 W electric power. This corresponds to $\sim 2\%$ power efficiency (from electric to optical power). Regarding optical amplifiers, the electric-to-optical power efficiency for our home-built EDFA was measured to be $\sim 3.4\%$ (50 mW output optical power for 1 mW input optical power from 1.46 W electric input power (1 W for pump diode current and 0.46 W for TEC)).

While the actual situations may vary greatly, here we will make simple scenarios to consider the overall power consumption issues when using this MLL comb and EDFAs for our photonic driverless CDNs. We will first consider the case for generating low-jitter clock signals for high-speed mixed-signal chips (such as transceivers or data converters). Given that the above-mentioned 2.5-GHz laser (powered by ~ 3 W electric power) can generate ~ 58 mW optical power, and assuming the photodiode responsivity of 0.9 A/W, the use of balanced photodiodes can generate a total photocurrent pulse of ~ 10 pC (for each rising and falling edges). When driving at 2.5-GHz clock rate (assuming $V_{dd} = 1$ V), this photocurrent pulse charge corresponds to driving clock loads of up to ~ 10 pF, which is capable of driving multiple transceivers or data converters operating at multi-GHz clock rates.

We can compare this electric power consumption to the scenario of providing low-jitter digital clock waveforms to data converters using a high-performance clock generating chip, even if it is not an exact 1:1 comparison situation. The AD9525, for instance, can generate eight clock signals with 63 fs jitter (by applying a separate external 3-GHz VCO as a reference) with the chip power consumption of ~ 1.2 W [AD9525 datasheet]. As a result, generating a low-jitter electronic clock signal from a PLL chip also uses watt-level power and will further have increased jitter and power consumption for clock distribution in the targeted chips. While our approach is not more energy efficient than electronic low-jitter clock generation methods, it is not considerably worse either (i.e., it will take watt-level electric power either photonic or electronic way for generating and distributing low-jitter clock signals), and it also offers unique benefits such as significantly lower jitter, skew, and heat dissipation in the chip as demonstrated in this work.

We also believe that our method may be better suited for scalable clocking of many chips in data center-like situations by using multiple fiber link branches and EDFAs (or EDWAs). For example, the total optical power of ~ 1.5 W (requiring 46.8 W electric power) can drive a total clock load of 700 pF at 1-GHz. Considering the clock load of a typical D flip-flop (in 65-nm process) is ~ 0.6 fF (which is determined from the simulation of our chip), this corresponds to more than a million flip-flops. With recent advances in chip-scale micro-combs and EDWAs (Liu et al, *Science* **376**, 1309 (2022)), the microcomb-plus-many-EDWAs may be a feasible option in the near future for distributing optical pulse signals to many CMOS or electronic-photonic hybrid chips with a compact and more power-efficient platform.

Revisions made: We added a paragraph on this issue in Discussion section (in pages 11-12; underlined red fonts): “Finally, we would like to briefly discuss on the overall electric power consumption and efficiency issue. As an example, for a 2.5-GHz repetition-rate laser, which is similar to the 2-GHz laser used in our work, it was recently reported that ~58 mW of output optical power was obtained with ~2 W electric power applied to the pump diode plus ~1 W electric power for thermoelectric cooling⁴⁵. The total extractable photocurrent pulse charge (~10 pC) from the given optical power is capable of driving multiple transceivers or data converters. While our method may not be more energy efficient than electronic clock generation methods (for example, ~1.2 W power consumption for the case of AD9525 chip⁴⁶), it is not considerably worse either, and it also offers unique benefits such as significantly lower jitter, skew, and heat dissipation in the chip. Furthermore, by utilising multiple fibre link branches and EDFAs (or Er-doped waveguide amplifiers (EDWAs)⁴⁷), our method may be better suited for clocking many chips in data center-like environments. With recent advances in chip-scale micro-combs and EDWAs, the microcomb-plus-many-EDWAs may be a compact and power-efficient option for distributing and injecting optical pulse signals to multiple chips in the near future.”

Point 2-6

The authors use external delay lines to multiply up the frequency comb rep rate. Is this a viable approach for eventual integration of a high rate clock source?

In this work, we used the repetition-rate multiplier to show the frequency-scalable property of the proposed method (from 250 MHz to 1 GHz). For higher repetition rate, there are several possible optical sources. For example, waveguide mode-locked lasers can be used for few GHz repetition rate, and in this work, a 2-GHz laser was used as an optical pulse source (where the measured data shown in Fig. 1b). We believe that microresonator-based Kerr combs can be a viable option for the generation of high repetition-rate clock signals, ranging from few GHz [for example, Suh et al, *Optica* **5**, 65 (2018) showed as low as 1.86-GHz repetition rate] to all the way up to tens GHz repetition-rate soliton pulses [Kippenberg et al, *Science* **332**, 555 (2011)]. We also recently showed that the timing jitter of a 22-GHz silica Kerr micro-combs can be in the few femtosecond regime [Jeong et al, *Optica* **7**, 1108 (2020)], which shows the potential of using such micro-combs for high-speed clock applications. Another nice feature of such micro-combs is that they can be made on an integrated photonic platform, and eventually reduce the total size and cost of this driverless CDN approach.

Revisions made: We added more discussion on the use of Kerr micro-combs as the viable options for higher repetition-rate clock source in Discussion section (in page 9, underlined red fonts): “For higher clock rates, we believe that microresonator-based Kerr combs³⁴ can be a viable option, ranging from few GHz³⁵ to tens GHz. We also recently found that the timing jitter of a 22-GHz silica Kerr micro-combs can be in the few-fs regime³⁶, which shows the potential of using such micro-combs for high-speed, low-jitter clocking applications. Another nice feature of such micro-combs is that they can be made on an integrated photonic platform, and eventually reduce the total size and cost.”

Point 2-7

My general feeling is that the demonstration is effective, but that the relevance of this approach to actual implementation of optical clock distribution still leaves a number of questions open, that would benefit from the authors' comment in the manuscript, and that define the context in which the significance of the current work can be better evaluated.

We would like to thank Reviewer 2 once again for taking the time to provide such a careful analysis of the manuscript. We hope that our response letter adequately addressed all of the reviewers' concerns.

REVIEWERS' COMMENTS

Reviewer #1 (Remarks to the Author):

The authors have satisfactorily replied to all comments raised by the reviewers. It can be published as is.

Reviewer #2 (Remarks to the Author):

The authors have adequately addressed all of the concerns raised in my review, and that of the other reviewer, and have added comments into the manuscript on key points regarding power and photodiodes. Although the demonstration is with off-chip photodiodes, I think the authors have adequately addressed the possibilities of on-chip photodiode integration, and the advance presented in the manuscript can stand on its own, given that the focus is on the power and timing improvements. The added figure with chip-on-board results adds value in terms information on steps toward closer integration/co-packaging for lower replate clocks.

I think the paper represents an excellent result that will probably stimulate further work. I would support its publication in Nature Communications.

Point-by-point response to reviewers' comments

Reviewer 1

The authors have satisfactorily replied to all comments raised by the reviewers. It can be published as is.

We again thank the reviewer's constructive comments which greatly improved the quality of the paper.

Reviewer 2

The authors have adequately addressed all of the concerns raised in my review, and that of the other reviewer, and have added comments into the manuscript on key points regarding power and photodiodes. Although the demonstration is with off-chip photodiodes, I think the authors have adequately addressed the possibilities of on-chip photodiode integration, and the advance presented in the manuscript can stand on its own, given that the focus is on the power and timing improvements. The added figure with chip-on-board results adds value in terms information on steps toward closer integration/co-packaging for lower replate clocks.

I think the paper represents an excellent result that will probably stimulate further work. I would support its publication in Nature Communications.

We again thank the reviewer's constructive comments which greatly improved the quality of the paper.